# Effects of neoadjuvant stereotactic body radiotherapy plus adebrelimab and chemotherapy for triple-negative breast cancer: A pilot study

Guanglei Chen[1,2,3], Xi Gu[1,2,3†], Jinqi Xue[1,2,3†], Xu Zhang[1†], Xiaopeng Yu[1†], Yu Zhang[1,4], Ailin Li[1], Yi Zhao[1], Guijin He[1], Meiyue Tang[1], Fei Xing[1], Jianqiao Yin[1], Xiaobo Bian[1], Ye Han[1], Shuo Cao[1], Chao Liu[1,2,3], Xiaofan Jiang[1,2,3], Keliang Zhang[5], Yan Xia[6], Huajun Li[6], Nan Niu[1,2,3], Caigang Liu[1,2,3]*, On behalf of the Northeastern Clinical Research Alliance of Oncology (NCRAO)

[1]Department of Oncology, Shengjing Hospital of China Medical University, Shenyang, China; [2]Cancer Stem Cell and Translational Medicine Laboratory, Shengjing Hospital of China Medical University, Shenyang, China; [3]Innovative Cancer Drug Research and Development Engineering Center of Liaoning Province, Shenyang, China; [4]Department of Gastrointestinal Surgery, Yantai Affiliated Hospital of Binzhou Medical University, Yantai, China; [5]Liaoning Center for Drug Evaluation and Inspection, Shenyang, China; [6]Jiangsu Hengrui Pharmaceuticals, Shanghai, China

*For correspondence:
angel-s205@163.com

†These authors contributed equally to this work

## Abstract

**Background:** Emerging data have supported the immunostimulatory role of radiotherapy, which could exert a synergistic effect with immune checkpoint inhibitors (ICIs). With proven effective but suboptimal effect of ICI and chemotherapy in triple-negative breast cancer (TNBC), we designed a pilot study to explore the efficacy and safety of neoadjuvant stereotactic body radiotherapy (SBRT) plus adebrelimab and chemotherapy in TNBC patients.

**Methods:** Treatment-naïve TNBC patients received two cycles of intravenous adebrelimab (20 mg/kg, every 3 weeks), and SBRT (24 Gy/3 f, every other day) started at the second cycle, then followed by six cycles of adebrelimab plus nab-paclitaxel (125 mg/m² on days 1 and 8) and carboplatin (area under the curve 6 mg/mL per min on day 1) every 3 weeks. The surgery was performed within 3–5 weeks after the end of neoadjuvant therapy. Primary endpoint was pathological complete response (pCR, ypT0/is ypN0). Secondary endpoints included objective response rate (ORR), residual cancer burden (RCB) 0-I, and safety.

**Results:** 13 patients were enrolled and received at least one dose of therapy. 10 (76.9%) patients completed SBRT and were included in efficacy analysis. 90% (9/10) of patients achieved pCR, both RCB 0-I and ORR reached 100% with three patients achieved complete remission. Adverse events (AEs) of all-grade and grade 3–4 occurred in 92.3% and 53.8%, respectively. One (7.7%) patient had treatment-related serious AEs. No radiation-related dermatitis or death occurred.

**Conclusions:** Adding SBRT to adebrelimab and neoadjuvant chemotherapy led to a substantial proportion of pCR with acceptable toxicities, supporting further exploration of this combination in TNBC patients.

**Funding:** None.

**Clinical trial number:** NCT05132790.

## Editor's evaluation

This study presents a valuable finding of a novel combinatory regimen that integrates immuno-therapy, radiotherapy, and chemotherapy in the current refractory triple-negative breast cancer. The evidence supporting the claims of the authors is solid, although inclusion of a larger number of patient samples and an animal model would have strengthened the study. The work will be of interest to Clinicians working on breast cancer.

## Introduction

Triple-negative breast cancer (TNBC), defined as estrogen receptor, progesterone receptor, and human epidermal growth factor receptor 2 (HER2) negative, accounts for approximately 10–20% of all breast cancers (*Costa et al., 2017*). Characterized by higher tumor mutation burden and more inten-sive tumor-infiltrating lymphocytes infiltration in tumor microenvironment (TME) (*Loi et al., 2019*), TNBC is more sensitive to immune checkpoint inhibitors (ICIs) than any other subtypes (*Keenan and Tolaney, 2020*). Nowadays, the combination with pembrolizumab (a PD-1 blockade) and standard chemotherapy is recommended for TNBC patients with stages II–III according to National Comprehen-sive Cancer Network (NCCN) guidelines (*Gradishar et al., 2022*). However, there is still around 40% of TNBC patients cannot achieve a pathological complete response (pCR) according to KEYNOTE-522 (*Schmid et al., 2020*) and IMpassion031 (*Mittendorf et al., 2020*) trials, which call for more promising strategies.

Radiotherapy (RT) is the most critical locoregional treatment in solid malignancies and approx-imately half of patients may receive radiotherapy during the entire treatment period (*Atun et al., 2015*; *Citrin, 2017*). Stereotactic body radiotherapy (SBRT), a novel technique with higher doses of radiation delivery to the tumor lesion in a smaller number of fractions, can shorten treatment time and reduce exposure to the surrounding tissues (*Chen et al., 2020*). SBRT has already been widely applied to advanced breast cancer as salvage treatment targeting to osseous and brain metastasis or other oligometastatic sites (*Nicosia et al., 2022*; *Viani et al., 2021*). Preoperative SBRT may be advan-tageous for downstaging the tumor to enable breast conservation and improving pCR rate (*Piras et al., 2023*) and multiple trials investigating neoadjuvant SBRT in various malignancies (*Holyoake et al., 2021*; *Kishi et al., 2020*; *Liu et al., 2022*; *Novikov et al., 2021*) are currently ongoing. In addi-tion, preclinical evidence suggested that SBRT has immunomodulatory properties by inducing cancer cells death to boost neoantigen-specific memory immune response and upregulate the expression of PD-L1 on tumor cells (*Deng et al., 2014*; *Han et al., 2022*; *Park et al., 2015*). Given these, it would be promising to combine SBRT with ICIs and chemotherapy under the neoadjuvant setting.

Therefore, we conducted a prospective pilot study to determine the efficacy and safety of neoad-juvant SBRT in combination with adebrelimab (SHR-1316), a potent selective PD-L1 inhibitor, plus nab-paclitaxel and carboplatin in patients with newly diagnosed early or locally advanced TNBC, to validate the feasibility of this novel regimen in neoadjuvant treatment of TNBC.

## Methods

### Patients

Previously untreated patients aged between 18 and 75 years, with histologically confirmed invasive TNBC (defined as negative estrogen receptor, progesterone receptor, and HER2 status by American Society of Clinical Oncology/College of American Pathologists guidelines) and pathological tumor size ≥2.0 cm in MRI assessment, were eligible for enrollment. Other inclusion criteria included Eastern Cooperative Oncology Group performance score of 0–1, adequate marrow, hepatic, renal, and cardiac function.

Key exclusion criteria included bilateral, inflammatory, or occult breast cancer; active or a history of autoimmune disease; use of glucocorticoids or other immunosuppressive therapy within 2 weeks before the first study dose; a history of interstitial pneumonia; active tuberculosis; pregnancy, lacta-tion, and refusal to use contraception.

### Study design and treatment

This study was performed in accordance with the Declaration of Helsinki and the Good Clinical Prac-tice guidelines. The trial protocol and all amendments of this single-arm, prospective study was

approved by the Institutional Review Board and Ethics Committee of Shengjing Hospital of China Medical University. Written informed consent was obtained from each patient before enrollment, and publish consents were obtained from all participated patients before submission.

All patients intravenously received 8 cycles of adebrelimab (20 mg/kg every 3 weeks). SBRT (24 Gy/3 f) targeted to breast lesion was started at the second cycle every other day, and 6 cycles of nab-paclitaxel (125 mg/m² on days 1 and 8) and carboplatin (area under the curve 6 mg/mL per min on day 1) was given every 3 weeks since the third cycle.

Patients who completed or discontinued the neoadjuvant treatment could undergo surgery. If disease progresses during the neoadjuvant phase, the patient either proceeds to surgery or receives alternative neoadjuvant therapy. Surgery was performed 3–5 weeks after the last dose of neoadjuvant therapy. Recommended surgery and adjuvant therapy were administered as per local guidelines or institutional standards.

### Outcomes

The primary endpoint was pCR rate in the breast and axillary lymph nodes (ypT0/is ypN0). Secondary endpoints included ORR before surgery (defined as the proportion of patients with complete or partial response according to the Response Evaluation Criteria in Solid Tumors [RECIST] version 1.1), the proportion of residual cancer burden (RCB) 0-I, and safety according to the Common Terminology Criteria for Adverse Events (CTCAE) version 5.0.

### Statistical analysis

Efficacy was assessed in the modified intention-to-treat population, which included patients who had undergone radiotherapy. Safety was evaluated in all recruited patients who received at least one dose of study drug. All statistical analyses were conducted using SAS 9.4 (North Carolina, USA). Continuous data are presented as mean and standard deviation, or mean and 95% confidence interval (CI).

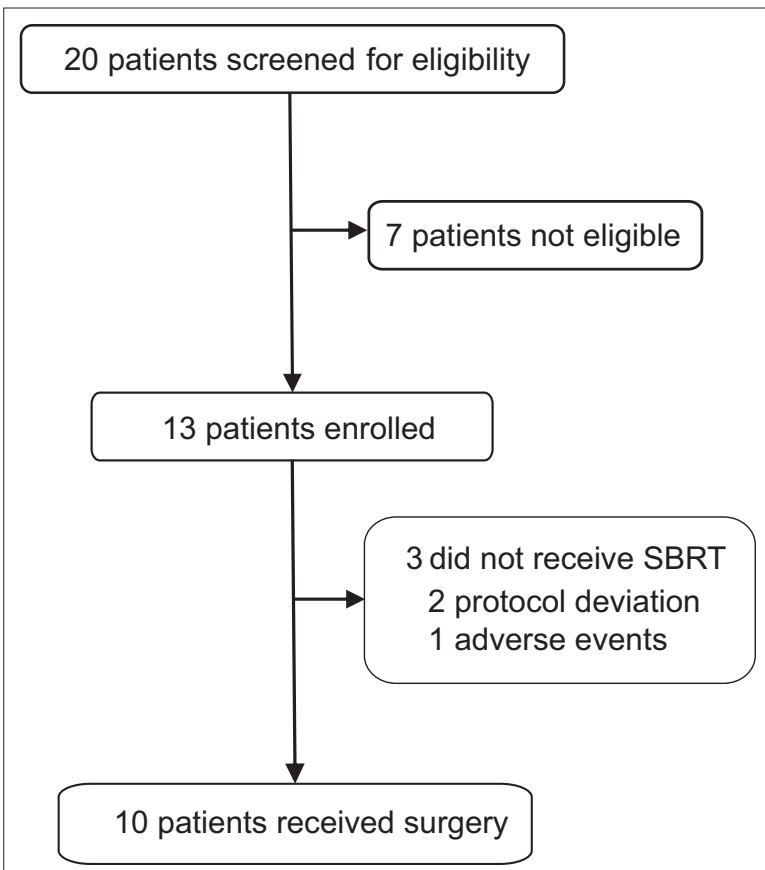

**Figure 1.** Flowchart of the trial. SBRT, stereotactic body radiotherapy.

**Table 1.** Baseline characteristics.

| Characteristic | Patients (n=13) |
| --- | --- |
| Age (median, range) | 51 (31–68) |
| Age group, years | |
| ≤50 | 6 (46.2%) |
| >50 | 7 (53.8%) |
| Menopausal status | |
| Premenopausal | 6 (46.2%) |
| Postmenopausal | 7 (53.8%) |
| Tumor size | |
| T2 | 10 (76.9%) |
| T3 | 3 (23.1%) |
| Lymph node status | |
| N0 | 6 (46.2%) |
| N1 | 2 (15.4%) |
| N2 | 5 (38.5%) |
| Clinical stage | |
| IIA | 6 (46.2%) |
| IIB | 1 (7.7%) |
| IIIA | 6 (46.2%) |
| Tumor grade | |
| II | 6 (46.2%) |
| III | 5 (38.5%) |
| Unknown | 2 (15.4%) |
| HER2 expression | |
| Negative | 8 (61.5%) |
| 1+ | 3 (23.1%) |
| 2+, FISH- | 2 (15.4%) |

Categorical data are expressed as frequency and percentage. The 95% CIs of pCR rate, proportion of patients with RCB 0-I, and ORR were estimated using the Clopper-Pearson method.

## Results
### Patient characteristics
Between December 2021 and January 2023, 13 patients were recruited in the trial (*Figure 1*). The baseline characteristics are shown in *Table 1*. The median age was 51 years (range 31–68) and the median tumor size was 33 mm (range, 22–72). Lymph node involvement was seen in 53.8% (7/13) patients and 46.2% (6/13) patients had stage III breast cancer at baseline.

### Outcomes
Among the 13 treated patients, 2 (15.4%) were excluded after the first dose of adebrelimab due to protocol deviation and 1 (7.7%) discontinued after the second dose of adebrelimab due to adverse event (AE), thus only 10 (76.9%) who underwent neoadjuvant SBRT and surgery (the modified intention-to-treat population) were available for efficacy evaluation (*Figure 1*). Nine of the 10 efficacy-evaluable patients (90%, 95% CI 55.5–99.8%) achieved pCR in the breast and axillary lymph nodes, and the rates of RCB 0-I was 100% (95% CI 67.9–100.0%). The radiological ORR was 100% (95% CI 67.9–100.0%, *Table 2* and *Figure 2*) with three patients achieved complete radiographic response. Four patients with positive lymph node at baseline had nodal downstaging to N0 after neoadjuvant treatment.

### Safety
AEs were reported in all 13 patients (*Table 3*). The incidence of any grade AEs was 92.3%. The grade 3 or higher treatment-related AEs occurred in 53.8% of the patients, including neutropenia (30.8%), anemia (7.7%), leukopenia (7.7%), thrombocytopenia (7.7%), creatine phosphokinase elevation (7.7%), and diarrhea (7.7%). Immune-related AEs of any grade occurred in 23.1% of the patients, including two (15.4%) patients had hyperthyroidism, one (7.7%) patient had hypothyroidism, and one (7.7%) patient experienced serious AEs due to immune-mediated myositis. Two patients required a dose reduction of carboplatin due to AEs but ultimately complete the prescribed treatment. There were no therapy-related death and no radiation-related dermatitis and skin hyperpigmentation.

## Discussion
This pioneering study reported the efficacy and safety of neoadjuvant SBRT in combination with adebrelimab, nab-paclitaxel, and carboplatin in TNBC patients. This new therapeutic regimen achieved promising anti-tumor activity with pCR rate of 90%, and the RCB 0-I and ORR rate of 100%, respectively. More importantly, this combinatory therapeutic strategy was well tolerated in this population.

**Table 2.** Pathological and clinical response.

| Variable | Patients (n=10) |
|---|---|
| Total pathological complete response | 9 (90%) |
| Residual cancer burden score | |
| 0 | 9 (90%) |
| I | 1 (10%) |
| II | 0 |
| III | 0 |
| Radiological response | |
| Complete response | 3 (30%) |
| Partial response | 7 (70%) |
| Stable disease | 0 |
| Objective response rate | 10 (100%) |

For TNBC patients who are candidates for preoperative therapy, neoadjuvant chemotherapy combined with ICIs has already achieved pCR rate of 58–64.8% in KEYNOTE-522 (*Schmid et al., 2020*) and IMpassion031 study (*Mittendorf et al., 2020*). In our study, the addition of SBRT to adebrelimab and standard neoadjuvant chemotherapy achieved a significantly higher percentage of pCR (90%). Since the dose of SBRT at 24 Gy in 3 fractions (bioequivalent dose=43.2 Gy) was lower than conventional preoperative radiotherapy dose (45–50 Gy/23–25 fractions) (*Ahmed et al., 2021*), we thought that SBRT (24 Gy/3 f) alone could rarely achieve pCR. We speculated that the SBRT (24 Gy/3 f) may exert synergy with immunochemotherapy, and not just local tumor eradication effects.

Recently, SBRT has been preclinically identified as exerting immunomodulatory effects and synergizing anticancer immune responses combined with ICIs (*Deng et al., 2014*; *Park et al., 2015*). *Pilones et al., 2020*, discovered that SBRT (two doses of 12 Gy) improved the therapeutic effects of PD-1 in a TNBC murine model, and this effect was enhanced by the addition of chemotherapy. TONIC trial demonstrated that SBRT (24 Gy/3 f) combined with nivolumab can increase the proportion of patients free of progression at 24 weeks than nivolumab alone (17% vs 8%) (*Voorwerk et al., 2019*). Another small sample research has indicated that preoperative SBRT (19.5–31.5 Gy/3 f) with neoadjuvant chemotherapy may result in fair pCR rates of 36%, with the maximum response (pCR 67%) was obtained at a dose of 25.5 Gy/3 f (*Bondiau et al., 2013*), as well as no increase in the incidences of early or late-term AEs (*Piras et al., 2023*; *Takanen et al., 2022*). Together, these studies suggested that SBRT may have strong immunomodulatory effects, rendering a synergistic anti-tumor effect with immunotherapy and chemotherapy.

Another study exploring SBRT (24 Gy/3 f) in combination with pembrolizumab plus chemotherapy in a neoadjuvant setting for TNBC patients also achieved a promising pCR rate of 74% (PEARL) (*McArthur et al., 2022*). Noteworthy, PEARL study chose taxane-based chemotherapy in which 52% of patients received platinum, while our study used nab-paclitaxel and carboplatin-based regimens. It is well established that neoadjuvant chemotherapy containing platinum has shown an increase in pCR rates of approximately 15.1% in TNBC patients (*Poggio et al., 2018*). Additionally, the IMpassion 130 (*Emens et al., 2021*) and IMpassion 131 (*Miles et al., 2021*) studies have suggested that nab-paclitaxel combined with ICIs may exhibit greater efficacy than paclitaxel. Thus, it is hypothesized that the selection of a chemotherapy regimen may contribute to the difference of pCR rates between the PEARL and our study (74% vs 90%). Consequently, it is imperative to determine the optimal neoadjuvant chemotherapy partners for ICIs in TNBC population.

AEs observed in our study were generally consistent with the known safety profiles of neoadjuvant therapy for TNBC patients in KEYNOTE-522 (*Schmid et al., 2020*) and IMpassion031 (*Mittendorf et al., 2020*) trials. The addition of SBRT did not increase any grades or grade 3 or above AEs, which were 92.3% vs 99% and 53.8% vs 57–76.8%, respectively. Furthermore, consistent with the customary toxic effects observed in lung cancers with adebrelimab plus carboplatin-based chemotherapy

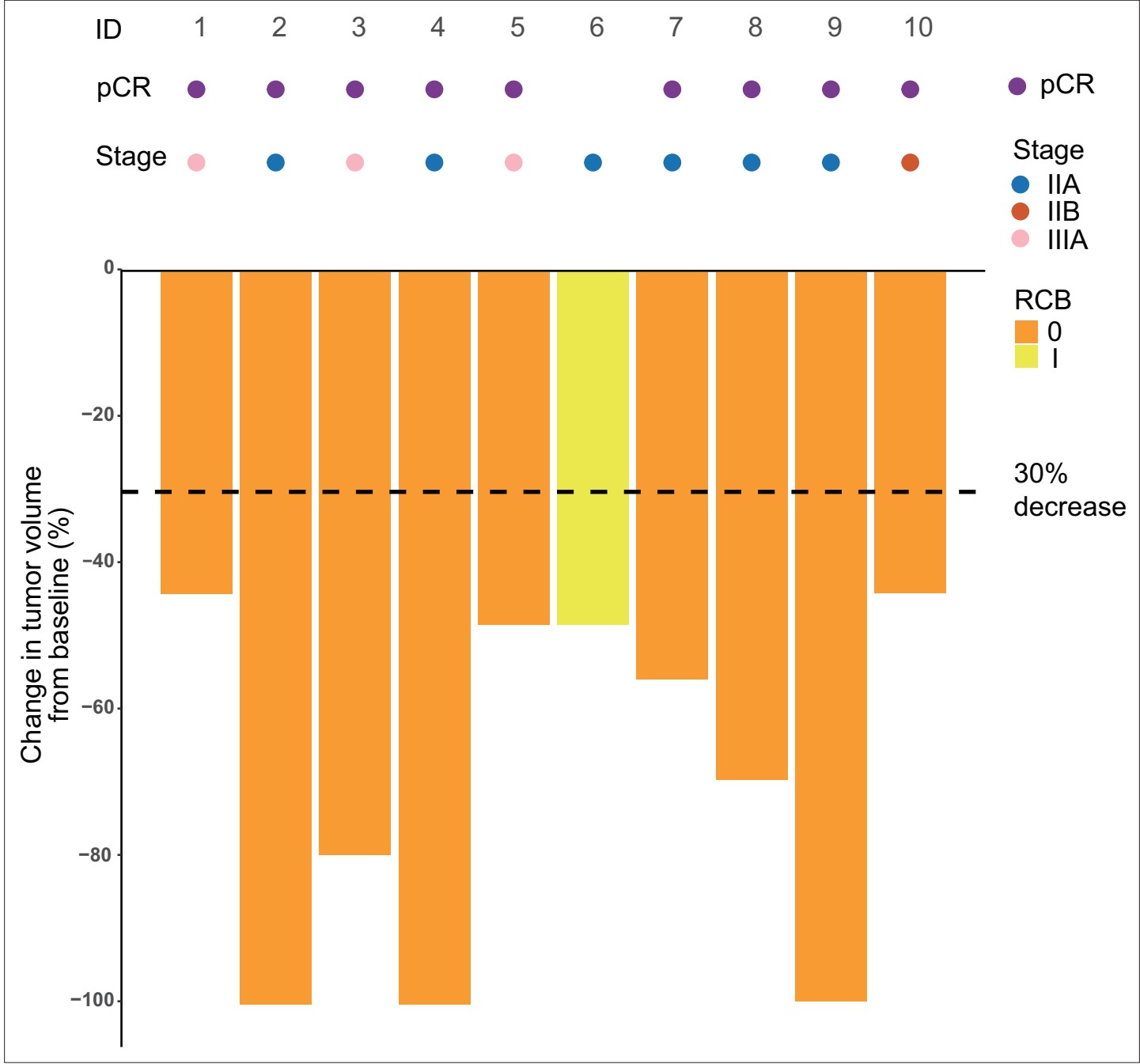

**Figure 2.** Swimming plot which demonstrates the pCR, RCB, and radiological response profile of 10 modified intention-to-treat population who received radiotherapy and undergone surgery. Each round dot or column indicates a patient. Colors indicate different clinical stage. ID, identity; pCR, pathological complete response; RCB, residual cancer burden.

The online version of this article includes the following source data for figure 2:

**Source code 1.** R Code for *Figure 2*.

**Source data 1.** Raw information for the pCR, RCB, and radiological response profile of 10 modified intention-to-treat population who received radiotherapy and underwent surgery.

**Table 3.** Treatment-related adverse events.

| | Patients (n=13) | | |
| | Grade 1 or 2 | Grade 3 | Grade 4 |
|---|---|---|---|
| Total | 5 (38.5%) | 6 (46.2%) | 1 (7.7%) |
| Anemia | 8 (61.5%) | 1 (7.7%) | 0 |
| Alopecia | 9 (69.2%) | 0 | 0 |
| Neutropenia | 4 (30.8%) | 3 (23.1%) | 1 (7.7%) |
| Hyponatremia | 8 (61.5%) | 0 | 0 |
| Nausea | 8 (61.5%) | 0 | 0 |
| Leukopenia | 6 (46.2%) | 1 (7.7%) | 0 |
| Lymphopenia | 7 (53.8%) | 0 | 0 |
| Thrombocytopenia | 5 (38.5%) | 1 (7.7%) | 0 |
| Elevated alanine aminotransferase level | 5 (38.5%) | 0 | 0 |
| Elevated aspartate aminotransferase level | 5 (38.5%) | 0 | 0 |
| Thyroid stimulating hormone decreased | 5 (38.5%) | 0 | 0 |
| Hyperuricemia | 5 (38.5%) | 0 | 0 |
| Vomiting | 5 (38.5%) | 0 | 0 |
| Fatigue | 4 (30.8%) | 0 | 0 |
| γ-Glutamyl transferase increased | 3 (23.1%) | 0 | 0 |
| Creatinine increased | 3 (23.1%) | 0 | 0 |
| Thyroid stimulating hormone increased | 3 (23.1%) | 0 | 0 |
| Creatine phosphokinase elevation | 1 (7.7%) | 1 (7.7%) | 0 |
| Hyperthyroidism | 2 (15.4%) | 0 | 0 |
| Free thyroid hormone decreased | 2 (15.4%) | 0 | 0 |
| Hyperglycemia | 2 (15.4%) | 0 | 0 |
| Hypokalemia | 2 (15.4%) | 0 | 0 |
| Rash | 2 (15.4%) | 0 | 0 |
| Peripheral neuropathy | 2 (15.4%) | 0 | 0 |
| Diarrhea | 0 | 1 (7.7%) | 0 |
| Immune mediated myositis | 1 (7.7%) | 0 | 0 |
| Ventricular extrasystole | 1 (7.7%) | 0 | 0 |
| Free thyroid hormone increased | 1 (7.7%) | 0 | 0 |
| Hypothyroidism | 1 (7.7%) | 0 | 0 |
| Troponin I elevated | 1 (7.7%) | 0 | 0 |
| Proteinuria | 1 (7.7%) | 0 | 0 |
| Constipation | 1 (7.7%) | 0 | 0 |

regimen (*Wang et al., 2022*; *Yan et al., 2023*), the addition of SBRT neither brought new AEs nor increased the incidences of grade 3 or higher or serious AEs in our population. Immune-related AEs occurred in three (23.1%) patients. The addition of SBRT did not increase immune-related AEs and severity compared to KEYNOTE-522.

Our study had several limitations including a non-comparative preliminary trial with relatively small sample size, thereby impeding the comparison of our data with historical data due to insufficient statistical power. Thus, further prospective randomized clinical trial required to validate these outcomes is currently in the planning stage.

In conclusion, the addition of SBRT to adebrelimab and neoadjuvant platinum-containing therapy showed the possibility of a convenient and feasible regimen for TNBC with promising efficacy and acceptable toxicities. Neoadjuvant radiotherapy may enhance the response to immunochemotherapy through activating TME. Further confirmation of these findings in large-scale study is currently underway.

## Acknowledgements

Jiangsu Hengrui Pharmaceuticals Co, Ltd provided the study drug adebrelimab free of charge for patients enrolled in the study. We thank the patients and their families involved in this study. This research did not receive any specific grant from funding agencies in the public, commercial, or not-for-profit sectors.

## Additional information

### Competing interests
Yan Xia: Yan Xia is the employee of Jiangsu Hengrui Pharmaceuticals Co., Ltd. No other potential conflicts of interest were reported. Huajun Li: Huajun Li is the employee of Jiangsu Hengrui Pharmaceuticals Co., Ltd. No other potential conflicts of interest were reported. Caigang Liu: Senior editor, eLife. The other authors declare that no competing interests exist.

### Funding

No external funding was received for this work.

### Author contributions

Guanglei Chen, Resources, Data curation, Software, Methodology; Xi Gu, Validation, Methodology; Jinqi Xue, Visualization, Methodology; Xu Zhang, Data curation, Validation; Xiaopeng Yu, Ailin Li, Resources, Methodology; Yu Zhang, Chao Liu, Software, Methodology; Yi Zhao, Guijin He, Meiyue Tang, Fei Xing, Jianqiao Yin, Xiaobo Bian, Ye Han, Resources; Shuo Cao, Yan Xia, Huajun Li, Methodology; Xiaofan Jiang, Writing - original draft; Keliang Zhang, Investigation; Nan Niu, Conceptualization, Resources, Data curation, Formal analysis, Supervision, Validation, Investigation, Methodology, Writing - original draft, Writing – review and editing; Caigang Liu, Conceptualization, Resources, Supervision, Validation, Investigation, Visualization, Methodology, Project administration, Writing – review and editing

### Author ORCIDs

Ye Han http://orcid.org/0000-0003-2929-8122
Caigang Liu https://orcid.org/0000-0003-2083-235X

### Ethics

Clinical trial registration NCT05132790.
Human subjects: The study was approved by Institutional Review Board of Shengjing Hospital, China Medical University and performed, according to the Declaration of Helsinki and Good Clinical Practice guidelines. Written informed consent was obtained from each patient.

### Decision letter and Author response
Decision letter https://doi.org/10.7554/eLife.91737.sa1
Author response https://doi.org/10.7554/eLife.91737.sa2

## Additional files

### Supplementary files
• MDAR checklist

• Reporting standard 1. The STROBE checklist.

### Data availability
The data are not publicly available due to privacy or ethical restrictions or laws and regulations, which are available from the principal investigator (liucg@sj-hospital.org) on reasonable request. Qualified researchers should submit a proposal to the principal investigator outlining the reasons for requiring the data. The PI and IRB affiliated to Shenging Hospital will check whether the request is subject to any intellectual property or confidentiality obligations. De-identified data will then be transferred to the inquiring investigator over secure file transfer. Commercial use of this data must comply with the requirements of Human Genetics Resources Administration of China and other country, a signed data access agreement with the principal investigator is required before commercial research be performed. Source data files have been provided for Figure 2. The code that was used to analyze the data has already been provided, shown in word document (Figure 2 - source data 1).

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
