## [Editor Report]

This study presents a valuable finding of a novel combinatory regimen that integrates immunotherapy, radiotherapy, and chemotherapy in the current refractory triple-negative breast cancer. The evidence supporting the claims of the authors is solid, although inclusion of a larger number of patient samples and an animal model would have strengthened the study. The work will be of interest to Clinicians working on breast cancer.

---

## [Decision Letter]

**Decision letter after peer review:**

Thank you for submitting your article "Efficacy and Safety of neoadjuvant stereotactic body radiotherapy plus adebrelimab and chemotherapy for triple-negative breast cancer: A pilot study" for consideration by eLife. Your article has been reviewed by 3 peer reviewers, one of whom is a member of our Board of Reviewing Editors, and the evaluation has been overseen by Mone Zaidi as the Senior Editor.

Essential revisions

*Reviewer #1 (Recommendations for the authors):*

1. This study investigated a novel neoadjuvant regimen in early triple negative breast cancer, and the primary endpoint was selected as the pCR rate. While overall survival is the golden parameter for evaluating cancer-related survival, and whether the improvement of pCR rate in this study could predict the long-term survival benefit should be addressed.

2. This clinical trial explored the efficacy and safety of neoadjuvant radiotherapy combined with immunotherapy and chemotherapy in early stage triple negative breast cancer. Please add the adverse effects of radiotherapy.

3. It will be better to modify the language by the English native speaker.

*Reviewer #2 (Recommendations for the authors):*

1. This work has innovatively and comprehensively demonstrated the efficacy and safety of immunotherapy which blocks PD-L1, combined with SBRT and chemotherapy in previous refractory triple-negative breast cancers. As side effects arise from SBRT and chemotherapy are commonly reported, whether the adverse events in this research are associated with specific immunotherapy should be explained, and strategies to deal with these AEs should be mentioned in the Discussion section.

2. The proper administration sequence of PD-L1 immunotherapy and radiotherapy should be supplemented, whether concurrent administration or sequential application of immunotherapy and radiotherapy could achieve better efficacy should be described.

*Reviewer #3 (Recommendations for the authors):*

1. Immunotherapy + chemotherapy is the standard treatment recommended by current guidelines for high-risk triple-negative breast cancer. In this study, radiation therapy was administered to the primary breast lesion, and radiation therapy can cause an abscopal effect, so it is recommended to supplement the regression of lymph nodes in the node positive patients after radiation therapy and at the time of surgery.

2. The relatively small sample size of this study is the major defect of this study, and further confirmatory studies are still needed.

---

## [Author Response]

Essential revisionsReviewer #1 (Recommendations for the authors):1. This study investigated a novel neoadjuvant regimen in early triple negative breast cancer, and the primary endpoint was selected as the pCR rate. While overall survival is the golden parameter for evaluating cancer-related survival, and whether the improvement of pCR rate in this study could predict the long-term survival benefit should be addressed.

It is really true as Reviewer#1 suggested that overall survival is the golden parameter. However, a pooled analysis of 12 neoadjuvant chemotherapy breast cancer clinical trials found that patients with breast cancer who achieved a pCR had significantly improved survival outcomes, and the association between pCR rates and long-term outcomes was strongest in patients with triple-negative breast cancer (OS: 0·16, 0·11–0·25). Considering that pCR rate is a surrogate endpoint for OS, we will continue to follow-up the patients and report the updated survival data.

References:

Pathological complete response and long-term clinical benefit in breast cancer: the CTNeoBC pooled analysis. Lancet. 2014

2. This clinical trial explored the efficacy and safety of neoadjuvant radiotherapy combined with immunotherapy and chemotherapy in early stage triple negative breast cancer. Please add the adverse effects of radiotherapy.

Thank you for your good suggestion. The adverse effects of radiotherapy were added both on Abstract (page 2, line 42) and Results section (page 8, line 154-155).

a) Abstract (page 2, line 42): No radiation-related dermatitis or death occurred.

b) Results section (page 8, line 154-155): There were no therapy-related death and no radiation-related dermatitis and skin hyperpigmentation.

3. It will be better to modify the language by the English native speaker.

We have polished the manuscript according to the Reviewer’s comments.

Reviewer #2 (Recommendations for the authors):1. This work has innovatively and comprehensively demonstrated the efficacy and safety of immunotherapy which blocks PD-L1, combined with SBRT and chemotherapy in previous refractory triple-negative breast cancers. As side effects arise from SBRT and chemotherapy are commonly reported, whether the adverse events in this research are associated with specific immunotherapy should be explained, and strategies to deal with these AEs should be mentioned in the Discussion section.

We have made amendments according to the Reviewer#2’s comments. The adverse effects of immunotherapy were added in Results section (page 8, line 151-152) and Discussion section (page 11, line 204-205).

a) Results section (page 8, line 151-152): Immune-related adverse events of any grade occurred in 23.1% of the patients, including 2 (15.4%) patients had hyperthyroidism, 1 (7.7%) patient had hypothyroidism, and 1 (7.7%) patient experienced serious adverse events due to immune-mediated myositis.

b) Discussion section (page 11, line 204-205): Immune-related adverse events occurred in 3 (23.1%) patients. The addition of SBRT did not increase immune-related adverse events and severity compared to KEYNOTE-522.

2. The proper administration sequence of PD-L1 immunotherapy and radiotherapy should be supplemented, whether concurrent administration or sequential application of immunotherapy and radiotherapy could achieve better efficacy should be described.

It is really true as Reviewer #2 suggested that whether concurrent administration or sequential application of immunotherapy and radiotherapy could achieve better efficacy. Data from mouse models have been obtained in support of various treatment schedules, encompassing radiotherapy immediately followed by ICIs, concomitant radiotherapy plus ICIs, as well as ICIs immediately followed by radiotherapy. The reason why we adopted immunotherapy induction administration was for preclinical research had reported that immunotherapy initiated before radiation has been shown to mediate superior therapeutic effects in mouse model.

References:

Emerging evidence for adapting radiotherapy to immunotherapy. Nat Rev Clin Oncol. 2023

Reviewer #3 (Recommendations for the authors):1. Immunotherapy + chemotherapy is the standard treatment recommended by current guidelines for high-risk triple-negative breast cancer. In this study, radiation therapy was administered to the primary breast lesion, and radiation therapy can cause an abscopal effect, so it is recommended to supplement the regression of lymph nodes in the node positive patients after radiation therapy and at the time of surgery.

As Reviewer #3 suggested, we have compared the regression of lymph nodes between baseline and after radiation therapy/at the time of surgery, respectively. There was no obvious regression in lymph nodes after radiation therapy, but all lymph nodes shrank to normal size at the time of surgery and confirmed pCR.

2. The relatively small sample size of this study is the major defect of this study, and further confirmatory studies are still needed.

Thank you for your good suggestion. It is really true as Reviewer #3 suggested that our study is a relatively small-scale trial. Further confirmation of these findings in a large-scale study is currently underway.